# Biomechanics Analysis of the Firefighters’ Thorax Movement on Personal Protective Equipment during Lifting Task Using Inertial Measurement Unit Motion Capture

**DOI:** 10.3390/ijerph192114232

**Published:** 2022-10-31

**Authors:** Muhamad Nurul Hisyam Yunus, Mohd Hafiidz Jaafar, Ahmad Sufril Azlan Mohamed, Nur Zaidi Azraai, Norhaniza Amil, Remy Md Zein

**Affiliations:** 1School of Industrial Technology, Universiti Sains Malaysia (USM), Penang 11800, Malaysia; 2School of Computer Sciences, Universiti Sains Malaysia (USM), Penang 11800, Malaysia; 3School of the Arts, Universiti Sains Malaysia (USM), Penang 11800, Malaysia; 4National Institute of Occupational Safety and Health (NIOSH), Bangi 43650, Malaysia

**Keywords:** angular kinematic, personal protective equipment, motion capture, ergonomic risk assessment, biomechanics

## Abstract

Back injury is a common musculoskeletal injury reported among firefighters (FFs) due to their nature of work and personal protective equipment (PPE). The nature of the work associated with heavy lifting tasks increases FFs’ risk of back injury. This study aimed to assess the biomechanics movement of FFs on personal protective equipment during a lifting task. A set of questionnaires was used to identify the prevalence of musculoskeletal pain experienced by FFs. Inertial measurement unit (IMU) motion capture was used in this study to record the body angle deviation and angular acceleration of FFs’ thorax extension. The descriptive analysis was used to analyze the relationship between the FFs’ age and body mass index with the FFs’ thorax movement during the lifting task with PPE and without PPE. Sixty-three percent of FFs reported lower back pain during work, based on the musculoskeletal pain questionnaire. The biomechanics analysis of thorax angle deviation and angular acceleration has shown that using FFs PPE significantly causes restricted movement and limited mobility for the FFs. As regards human factors, the FFs’ age influences the angle deviation while wearing PPE and FFs’ BMI influences the angular acceleration without wearing PPE during the lifting activity.

## 1. Introduction

Firefighting is one of the most strenuous jobs, involving many emergency and rescue operations [1,2]. Firefighters’ duties are to prevent and suppress a fire, rescue, provide medical care, and perform numerous responsibilities related to emergency service [1]. FFs need to be psychologically and physically fit to suit the nature of the job [3,4]. Various studies have reported injuries involving FFs, such as exposure to fire, chemicals or radiation, extreme weather, strain, fall, slip, and trip [2,5,6,7]. In order to reduce the risk that FFs are exposed to, they are equipped with PPE. The personal protective equipment consists of a coat, pants, boots, hood, helmet, and a self-contained breathing apparatus (SCBA) tank that acts as the last resort for safety standing between FFs and potential harm [8,9].

However, the bulkiness and heaviness of FF PPE have been reported to result in musculoskeletal disorders [8,9,10,11,12]. Musculoskeletal disorder is one of the most significant injuries related to FFs [6,11]. Their duties involve awkward postures, repetitive movement, heavy lifting, and overreaction, increasing the risk of musculoskeletal injuries [1,13,14]. FFs are exposed to musculoskeletal disorders during emergency operations and potentially during training and general tasks [13,15]. One of the significant injuries related to FF is a back injury associated with musculoskeletal injuries [1,15]. FFs are exposed to back injuries due to extreme jobs, which demand physical strength and heavy protective equipment [6,10]. Therefore, ergonomic studies for FFs are important to reduce the risk of musculoskeletal injuries.

Manual handling activities among FFs are one of the significant factors that cause musculoskeletal disorders [6,16]. A FF who performs manual handling activities such as lifting, pushing, and pulling might develop a back injury [16,17,18]. FFs do various heavy lifting such as lifting a fire hose, the SCBA tank, fire equipment, and victims during fire rescue operations. Heavy lifting tasks with unergonomic posture will cause back injury in FFs [19,20]. Therefore, this study selected a lifting task in order to analyze the kinematic data of a FF’s movements. 

This study will focus on two variables to discuss FF movement: body angle deviation and body angular acceleration. Body angle deviation in ergonomic fields is important to identify the correct body posture while doing a task [21,22]. In short, ergonomic body posture has less body angle deviation and less movement [20,23]. Body angular acceleration describes the rotational movement of body limbs [24]. Angular motion in biomechanics study is important as most body movements are angular motions of limbs and joints, which means joints act as the centripetal force of the limb. Figure 1 describes the angular motion of the human body movement.

Motion capture can be divided into optical and non-optical motion capture [25]. Both motion captures have been used widely in ergonomic and biomechanics studies [22,26]. The advantage of motion capture in biomechanics studies is that it can provide kinematic data and body skeletons that can help analyze the movement [22,27]. IMU is a non-optical motion capture that depends on accelerometers and gyroscope sensors [28,29]. 

Qualitative studies related to the restriction of movement and limited mobility of PPE have been extensively conducted, but quantitatively, not much has been discussed [10,28,29,30,31]. Therefore, this study attempts to apply body angle and angular acceleration data to discuss the comparison of FFs’ movement while wearing the full PPE and without wearing PPE, focusing on the FF’s thorax movement during the lifting task. Thorax is the body region between the neck and the abdomen that consists of ribs, breastbone, and spine. The kinematic data and the angle deviation of the thorax can help the study determine the cause of back injury.

## 2. Materials and Methods

### 2.1. Data Collection

Prior to initiating this research project, study protocols were approved by the Institutional Review Board at the University Sains Malaysia (Code No.: USM/JEPeM/19080456). The biomechanics study aimed to analyze the behavior of FFs’ movement during lifting tasks with and without PPE. Nineteen Malaysian FFs from Zone 1 Penang Firefighter Department participated in this study. The selection of FFs for participation in this study was based on inclusion and exclusion criteria. Only experienced FFs with a grade of KB19-26 and FFs who passed the Individual Physical Proficiency Test (IPPT) could participate in this study. KB19-26 is the lowest recognized FF grade in Malaysia. FFs diagnosed with critical diseases or injuries were excluded from the study.

The participants for this study were divided into five age groups: (I) below 25 years old; (II) 26 to 30 years old; (III) 31 to 35 years old; (IV) 36 to 40 years old; and (V) 41 years old and above. The total number of FFs at Zone 1 (Northwest) Penang is 201, with 9% from group I, 21% from group II, 24% from group III, 19% from group IV, and 27% from group V. The sample size of this study was calculated using the following formula.
(1)$$n=\frac{z^{s}P\left(1-P\right)}{\in^{2}}\;$$

The confidence level was set at 95%, while the z-score that was obtained was 1.96. The proportion (*P*) was assumed to be 0.5, and the margin of error (∈) was 22%. The total sample size needed for this study was 19 FFs that comprise group I (2 participants), group II (4 participants), group III (4 participants), group IV (4 participants), and group V (5 participants).

The musculoskeletal discomfort survey was used in this study to assess each of the FFs’ work experiences (Appendix A) [32]. From the musculoskeletal discomfort survey, the FFs were required to indicate the body part that experienced injury or discomfort, either annual prevalence or discomfort from work. This survey indicates the potential risk of body parts for this study. Apart from musculoskeletal discomfort, the FFs were also required to fill in their demographic background, consisting of age, year of service, height, and weight. As the survey collected the weight and height of FFs, this study also categorized the data collection into body mass index (BMI) groups to analyze the relationship between BMI and the FFs’ PPE towards the thorax extension during the lifting task. The BMI of FFs was divided into three groups, which are group (I): between 19.0 to 24.9; group (II): between 25.0 to 29.9; and group (III): 30.0 and above. 

The Rokoko Smartsuit Pro motion capture (Rokoko, Copenhagen, Denmark) was used in this study to collect the kinematic data of FFs’ movement consisting of body velocity and acceleration. The compatibility of Rokoko motion capture in biomechanics studies was tested by recent research [33]. Each FF was requested to wear a motion capture suit with 19 sensors attached to the different parts of their body. All the Rokoko Smartsuit Pro sensors were connected to a cable connected to a motherboard, as shown in Figure 2. The motherboard is a hub connecting to the computer through a WIFI connection provided by the WIFI router. A battery was used as a power supply to the hub and the sensor of the Rokoko Smartsuit. The Rokoko Smartsuit comprises a high-quality nylon fabric that is durable and elastic and that various body types can wear. The suit provided an adjustable strap to tighten the sensor to the body. This strap prevented the relative motion during the experiment and decreased the error. All 19 sensors and the hub were placed into the hidden pockets inside the suit. The wire that connected the sensor and the hub was also hidden throughout the suit’s integrated tunnel. This tunnel protects the wire and the sensor without restricting the motion during the experiment. Figure 2 shows the Rokoko Smarstuit hardware setup. 

The connection through the Wi-Fi can transfer the data within a 100 m distance. The data transfer from the suit is done in real-time, with a transfer rate of 100 frames per second. As the primary connection of the hub is Wi-Fi, any magnetic interference can cause errors in the data collection. Thus, creating a magnetic safe zone was required during the experiment. Materials such as steel and iron, and devices that potentially create local magnetic fields were avoided during the experiment. The FFs’ SCBA tank was duplicated using non-magnetic materials to reduce data collection errors.

The task selection was based on a structured interview with FF personnel regarding the ergonomically challenging tasks that frequently cause musculoskeletal discomfort among FFs. The selected activity, the fire hose ground lifting task, requires thorax extension performance, as shown in Figure 3. This activity was identified as having a high potential to develop musculoskeletal discomfort in the back region [34,35,36]. Each FF was requested to perform the lifting task wearing PPE and without wearing PPE. This will help to understand the effect of FFs’ PPE on the FF’s movement.

### 2.2. Data Extraction

Biomechanically, human movement has four main types: flexion, extension, abduction, and adduction [33,37]. Extension describes the body movement that involves anterior or posterior movement within the sagittal plane of the human body [37]. The body deviation angle is when the body deviates from the neutral position [37]. This study focuses on the angle deviation and angular acceleration of the thorax extension of FFs, as shown in Figure 4. The body deviation angle is important to quantify the awkward posture. An awkward posture is one of the ergonomic risk factors that can lead to musculoskeletal disorders [23].

The Rokoko motion capture creates a line of the vector between two respective sensors to calculate the angle of deviation. The unit for the angular acceleration used in this study is the degree (deg). Figure 4 describes the example of a vector line to calculate the angle of deviation. Vector $\vec{A}$ is the stationary position, while vector $\vec{B}$ is the point during the movement. From the vector $\vec{A}$ and $\vec{B}$, the angle deviation is calculated using the arctangent trigonometric function as shown in Equations (2)–(6).
(2)$$\vec{OA}=\left(x_{a}-x_{0},y_{a}-y_{o},z_{a}-z_{o}\right)$$
(3)$$\vec{OB}=\left(x_{b}-x_{0},y_{b}-y_{o},z_{b}-z_{o}\right)$$
(4)$$\vec{AB}=\left|OA\right|\left|OB\right|\cos\alpha$$
(5)$$\vec{A}\times \vec{B}=\left|OA\right|\left|OB\right|\;\sin\;\alpha$$
(6)$$\tan\alpha =\frac{\vec{A}\times \vec{B}}{\vec{A}\cdot \vec{B}}$$

This study will present angular acceleration to describe the motion of FFs. Angular acceleration is important in the ergonomic study to quantify the change of speed for every movement point [24]. Acceleration is a vector variable that describes the magnitude and direction of motion. Angular acceleration can be defined as the derivative of angular velocity, as shown in Equation (7).
(7)$$\mathrm{Angular}\;\mathrm{acceleration}\;\left(\alpha \right)=\frac{d^{2}\theta }{dt^{2}}=\frac{d\omega }{dt}$$
where θ is angular displacement and ω is angular velocity. The unit for the angular acceleration used in this study is $\deg/s^{2}$ and the data were captured at the rate of 100 frames per second.

In short, this study discusses the angle deviation and angular acceleration of the thorax movement of FFs during a lifting task. The angle deviation describes the posture, while the angular acceleration describes the thorax motion of the FFs [38].

### 2.3. Data Analysis

The questionnaire survey consisted of 19 male FFs aged 21 to 53. Their average age, height, and BMI were 35.7, 170.8 cm, and 24.5. Thirty-seven percent of the respondents had less than 10 years of service. All of them were full-time and trained FFs. The demographic background of FFs involved in this study is described in Table 1.

The IBM SPSS Statistic for windows, version 26.0 (IBM Corp., Armonk, NY, USA) was used for data analysis. Descriptive statistics (Means M), standard deviations (SD), maximum value, and percentages were used to describe the angle deviation and angular acceleration of thorax extension collected by the motion capture. The ANOVA statistical test was used to analyze the comparison between the age and BMI groups. The *t*-test was used to analyze the comparison of the angle deviation and angular acceleration of the thorax between those without PPE and those with PPE. 

## 3. Result

Based on the data collected from the musculoskeletal discomfort survey shown in Figure 5, most respondents reported pain in certain parts of their bodies, with 16% and 32% not reporting any pain from the annual prevalence and work-related. The most-reported pain was lower back, with 63% of the subjects who reported lower back pain agreeing that it is work-related. Other body parts that show high reports among FFs experiencing discomfort are the neck (32%), shoulder (21%), and upper back (21%), with subjects agreeing that the pain is work-related. On the other hand, 21% and 26% of respondents reported an annual prevalence of musculoskeletal discomfort in the left and right knee, respectively. Furthermore, 26% of FFs experience an annual prevalence of musculoskeletal pain, and 21% agree that the pain is work-related for the right wrist. As regards the left wrist, only 11% of FFs reported an annual prevalence of pain, with 5% agreeing the pain is work-related.

Figure 6 shows the thorax angle deviation and angular acceleration collected by motion capture. According to Figure 5, 74% of FFs reduce the angle deviation of thorax extension after wearing the PPE to complete the same hose-lifting task. While only 1 of the FFs does not show any changes in angle deviation after wearing the PPE, 16% of FFs show a minimally increasing angle deviation. FFs P18 and P19 showed the most significant differences in thorax angle deviation after wearing the PPE compared to other participants. Notably, FFs P18 and P19 were the oldest participants at 46 and 53 years old. The figure also shows that 68% of FFs reduce thorax angular acceleration during lifting tasks after wearing the PPE. Only P8 recorded an outstanding increase in angular acceleration after wearing the PPE with 3889.58 deg/s^2^. FFs that recorded a higher angular acceleration are P19 with 6599.24 deg/s^2^ followed by P18 with 558.51 deg/s^2^. However, both FFs recorded a remarkable reduction of angular acceleration after wearing PPE with 2072.88 deg/s^2^ and 383.81 deg/s^2^, respectively.

Table 2 shows the result of the *t*-test and ANOVA. The table shows that the angle deviation between PPE and without PPE conditions during lifting activity is the most significant, with a *p*-value of 0.001. It indicates that using PPE significantly influences the angle deviation during lifting. The ANOVA test shows a significant comparison between the age group of FFs for angle deviation of the thorax while wearing PPE with a *p*-value of 0.045. The ANOVA test also shows a significant comparison between the BMI group of FFs for angular acceleration without PPE with a *p*-value of 0.047.

Figure 7 shows the angle deviation and angular acceleration based on the age group and BMI group of FFs. According to the figure, all the age and BMI groups recorded different maximum angles and angular acceleration during the lifting activity. Most of the group shows a reduced angle and angular acceleration after wearing PPE.

Figure 7a shows an increasing angle deviation from G2 to G5 during the lifting task without PPE. However, the youngest age group (G1) recorded a higher angle deviation than G2 and G3. The figure shows that G5, the oldest age group, recorded the highest maximum angular acceleration. Figure 7c shows that the thorax angle deviation of lifting without PPE is decreasing across the BMI group while increasing for lifting with PPE. The angle deviation differences between lifting with PPE and without PPE get smaller as the BMI increase. BMI of G3 shows the smallest angle difference, with 52.04 deg for lifting without PPE and 42.30 deg for lifting with PPE. According to Figure 7d, the trend for the average maximum angular for lifting without PPE is increasing from G1 to G2 and decreasing for G3, while lifting with PPE is increasing across the BMI group. The standard deviation for lifting without PPE is too high for G1 and G2, with 1524.68 deg/s^2^ and 1948.12 deg/s^2^, respectively. The high standard deviation is because of the age factor influencing their physical strength.

## 4. Discussion

This chapter will discuss the data collection, consisting of musculoskeletal discomfort among FFs, FFs’ lifting posture, and the effect of PPE on the FFs’ lifting motion.

### 4.1. Musculoskeletal Discomfort among FFs

According to the musculoskeletal discomfort survey, the most reported musculoskeletal discomfort of the body parts among FFs is the back body region. It shows that the various lifting tasks performed by the FFs and the use of PPE may be the primary cause of lower back pain. Therefore, this study focused on the movement of the thorax as it will describe the movement of the back region of the body.

Three factors cause musculoskeletal discomfort among FFs: the human factor, the work factor, and the use of PPE. From the results collected, 10 FFs were categorized as heavyweight, and two as obese. A high BMI is a high risk factor of low back pain [39]. This is illustrated by the musculoskeletal discomfort survey, in which 63% of FFs reported lower back discomfort. 

The work factor can be explained by the work nature of FFs, which consists of heavy equipment and a lot of manual handling. Manual handling is recognized as an occupational risk that can potentially cause musculoskeletal disorders [40]. Therefore, this research will study the lifting activity as a part of manual handling performed by FFs.

In addition, using PPE can cause musculoskeletal discomfort to the FFs [29,30]. The FFs’ PPE has been reported to cause restriction of movement for FFs [10]. An example of issues related to the FFs’ PPE is the bulkiness and the poor size. The bulkiness of PPE can be associated with the weight of the SCBA tank used by FFs. The SCBA tank will increase the load on the back of FFs, and will, thus, cause musculoskeletal discomfort in the long term. It can be seen from the result of the musculoskeletal discomfort survey which low back injuries are the most reported cases by FFs. The risk of FFs developing musculoskeletal discomfort in the ankle area due to unfit and poor-size boots is high [9,10,29]. The poor size will create an unfavorable movement that can cause sprain and strain. 

### 4.2. FFs Lifting Posture

In the motion capture experiment, two types of lifting postures were performed by 19 FFs and they are squatting and stooping postures, as shown in Figure 3. The body posture during lifting will directly affect the FF in developing a musculoskeletal injury [41,42,43]. Some postures are not suitable for heavy lifting. The fire hose used in this study is around 11 kg, and the weight can be doubled if the hose is in a wet condition. A squatting posture is more suitable than a stooping posture for heavy lifting [44]. The stooping posture consists of minimum body movement and only involves the upper part of the body, compared to the squatting posture, which involves whole-body movement. However, a stooping posture will place all the load and force during lifting on the spine. It will cause muscle fatigue in the back region in the long run [11]. Stooping postures are suitable for lifting tasks that involve a light object that requires less force [44]. 

Figure 8 shows the angle deviation and angular acceleration of thorax extension. The negative angle value indicates that the thorax is bending forward or in flexion. The negative angular acceleration value shows that the thorax movement slows down or the velocity decreases. Figure 8 shows that the angle deviation of the thorax for the stooping posture is higher than the squatting lifting posture at 130 deg and 30 deg, respectively. High body angle deviation will cause an awkward posture, increasing the risk of musculoskeletal injury [23,45]. An awkward posture is an ergonomic risk factor that should be avoided by performing body movements with a proper posture. The results from the angle deviation for both lifting postures show that lifting with a squatting posture can decrease the angle deviation and reduce the risk of injury.

Figure 7 shows the angular acceleration of thorax extension and a significant difference between both postures. The maximum acceleration recorded for the squatting lifting posture is 456 deg/s^2^, much lower than the stooping lifting posture with 900 deg/s^2^. The higher the acceleration, the higher the force, as force is directly proportional to acceleration. High force performed by the body will result in muscle fatigue. It is because performing a forceful exertion task will require considerable muscle contraction, which causes them to fatigue [6,46]. The more force applied during the task, the higher the risk for the muscle to fatigue or strain [46]. Prolonged exposure to the ergonomic risk factor of forceful exertion can cause musculoskeletal injury [47]. 

The stooping lifting posture also shows a high decreasing and increasing angular acceleration during the thorax bending downward and moving upward, as shown in Figure 8b. The significant change in acceleration shows the movement has a significant change in velocity that indicate high kinetic energy. In classical physic, kinetic energy is the energy created due to its motion and, thus, high velocity will develop high kinetic energy. As shown in Figure 8a, the difference between the maximum and minimum acceleration is slight, indicating the movement requires less kinetic energy than the angular acceleration of the squatting posture. This matter can be quantified by calculating the thorax angular acceleration’s average and sample standard deviation. A high standard deviation value shows that the body is performing high decreasing or increasing acceleration that develops high kinetic energy. As shown in Figure 9, the standard deviation for the stooping posture is relatively high compared to squatting. Therefore, proper posture during a lifting task is important to reduce the risk of musculoskeletal injury.

### 4.3. The Effect of PPE on the FFs Lifting Motion

Apart from body posture, the PPE used by the FFs also contributes to the movement of thorax extension. The reduction of angle deviation after wearing the PPE shows a factor of movement restriction experienced by the FFs. Many factors of PPE contribute to movement restriction, such as poor fitting and bulkiness [10,39]. The poor fit of PPE, especially in the joint area, will cause a limited range of motion of the joint, leading to poor dexterity. The poor dexterity from wearing PPE forced the FFs to reduce their movement speed while lifting the hose. The reduction of acceleration causes the lifting process to take longer than usual. The weight of the SCBA tank used in this study is 12 kg. To reduce the perceived heaviness of the SCBA tank, the FFs tend to reduce their mobility by performing minor thorax angle deviation [31,48]. This can be seen in that most of the participants recorded a reduction of the angular acceleration of the thorax after wearing the PPE. The reduction in acceleration indicates that the FFs’ movement is slowing down because of movement restriction and limited mobility. It shows that the FFs’ PPE can reduce their mobility, which can cause musculoskeletal injury. 

FFs who performed a stooping posture recorded an angle of more than 50  deg. However, most showed a significant decrease in angle after wearing the PPE. It is because they change their posture to squatting or semi-squatting. This change in posture occurs due to discomfort or load on their back due to wearing the SCBA tank, which causes them to change their posture during lifting to reduce the perceived heaviness [49]. The unfit and bulkiness of the PPE also might be a factor that causes the FFs to change their posture [10].

Among all FFs, P8 shows a significant increase in angular acceleration after wearing the PPE. P8 increases the lifting speed while wearing the PPE to reduce the perceived heaviness from the weight of the hose and the PPE. P18 and P19 show a high angular acceleration compared to other FFs during the lifting process without wearing the PPE. As mentioned before, P19 and P18 are the oldest FFs in this study. The age factor can contribute to the speed behavior of the FFs; the older a person, the less strength he has [50,51]. The strength of the individual and the load lifted will affect the speed of the FFs’ movement [50,52]. To reduce the perceived heaviness, the FFs will perform the lifting process faster at a speed that can be tolerated. Due to age, P19 and P18 might perform the lifting movement faster than other FFs to reduce the perceived heaviness. This is explained in Figure 6, which shows that the oldest age group (G5) recorded the higher average angular acceleration with 2890.21 deg/s^2^. Subjects with lower physical strength tend to increase their lifting pace to reduce their spinal loading [33,53,54]. The increasing lifting pace will produce a high angular acceleration of thorax extension. High angular acceleration indicates the high force exerted by the firefighters.

Besides the age factor, the effect of PPE on the thorax angle deviation and angular acceleration of FFs can also be seen from the BMI factors. From the result shown in Figure 7c, BMI for G3 is less affected by the use of PPE. It indicates that the higher the BMI, the less the effect of PPE on their lifting posture. The result also shows that the average maximum angular acceleration for BMI group G3 during the lifting without PPE is the lowest compared to the BMI group G1 and G2. Muscular strength increased with an increase in BMI [55]. High muscle strength tends to lift at a slow pace compared to low muscle strength as they can exert longer spinal loading [35]. However, BMI group G3 recorded the highest average of maximum thorax angular acceleration. The BMI group G3 has a high standard deviation with 20,498.87 deg/s^2^, which cause the average maximum thorax angular acceleration to be high. 

The movement speed of FFs during the lifting task will affect the back compressive force [35,53]. A slow lifting speed can be more hazardous than a fast-lifting speed as the lifting speed will increase the duration of dynamic spinal loading. The longer the spinal exposure to the compressive force, the higher the risk of back injury. In addition, the lifting speed will alter the temporal muscle recruitment pattern [54]. A faster lifting speed has high muscle recruitment in the initial phase but has low muscle recruitment demand in the terminal phase. Conversely, the slower lifting speed initially has a low muscle recruitment demand, and the muscle recruitment demand increases at the terminal phase. The increasing demand at the terminal phase causes a burden to the back body region and can contribute to a back injury. It shows that the application of PPE among FFs can indirectly cause back injuries.

Reducing equipment weight and redesigning PPE to reduce the risk of back injury is seen as an ideal intervention. However, the ergonomic intervention controlling lifting speed is more practicable and less expensive. Redesigning the FF PPE and modifying the equipment have relatively high costs and are time-consuming. Thus, FFs need to revise their working pace while wearing PPE. 

Figure 10 shows the relationship between PPE, human, and work factors in developing the ergonomic risk factor. The intervention steps from every factor to reduce the ergonomic risk are very useful for the FF. The angle deviation and angular acceleration of thorax extension during lifting tasks have shown that FFs’ PPE potentially increases the risk of back injury. 

The PPE causing bulkiness and poor fit will affect the comfort and mobility experienced by FFs. Decreasing comfort and mobility while wearing the PPE can reduce the functional performance of FFs as a measure of their work pace, range of motion, strain, and sprain injury. Reducing mobility by wearing PPE also will increase the risk of slip, trip, and fall during operation. The work nature of FFs, which is associated with heavy equipment, unsafe working posture, working pace, and manual handling, also potentially causes back injury. The ergonomic risk from the work factor can be reduced by assessing FFs’ working posture and pace. The study shows that it is essential for FFs to perform a full-squatting posture and reduce the lifting loading time by increasing the speed to decrease the back compressive force. Finally, the risk from the human factor can be avoided by recruiting FFs with good physical strength and in an appropriate age range.

## 5. Conclusions

The personal protective equipment of FFs exposes the FF to the risk of back injury. The working nature of FFs exposed to the various heavy tasks, which is associated with heavy lifting, high working pace, and unsafe work posture, increases the risk of back injury among FFs. In the study, the ergonomic assessment was conducted using the IMU motion capture to improve assessment accuracy. By applying motion capture, the kinematic data and the body angle deviation can be recorded digitally, reducing the data collection error. 

The musculoskeletal pain questionnaire identified that 63% of FFs had experienced lower back pain during work. The questionnaire also showed that the neck, shoulder, and upper back are the body parts that recorded pain among FFs, indicating that the back region is the most affected.

The study results show that body angle deviation and angular acceleration recorded by IMU motion capture are useful in assessing the movement and mobility of FFs during the lifting task. The study shows that using PPE during lifting significantly contributed to movement restriction and limited mobility among FFs. During the lifting task, the most affected part of the PPE to the body movement is the SCBA tank.

The limitation of the study is that this research cannot conclude the best practice of angular acceleration of thorax extension. Further study on the force applied by the subject to lift the load is needed. Study on muscle contraction during lifting should also be considered. The medical point of view related to the thorax movement is also needed to determine the best practice of thorax angular acceleration during lifting. 

For future studies, it is essential to quantify the force from the FFs’ movement for the ergonomic risk assessment. Calculating the force exerted by the FFs while performing their activities can help determine the risk level that FFs are exposed to. One of the ergonomic risk factors is forceful exertion, which cannot be evaluated by observing the posture movement. Next, future studies can conduct research related to the BMI factor with a suitable range of age to reduce the error and avoid external factors that influence the result.

In general, the result of this study showed that ergonomic intervention should be applied to improve safety among FFs. Ergonomic interventions related to the human, working, and PPE factors will significantly minimize the risk of back injury.

## Figures and Tables

**Figure 1 ijerph-19-14232-f001:**
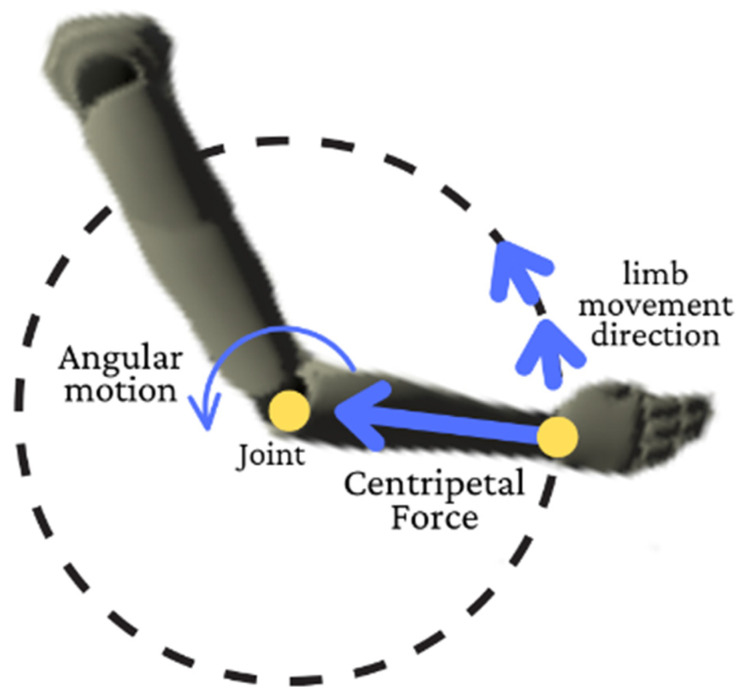
Angular motion of a human joint.

**Figure 2 ijerph-19-14232-f002:**
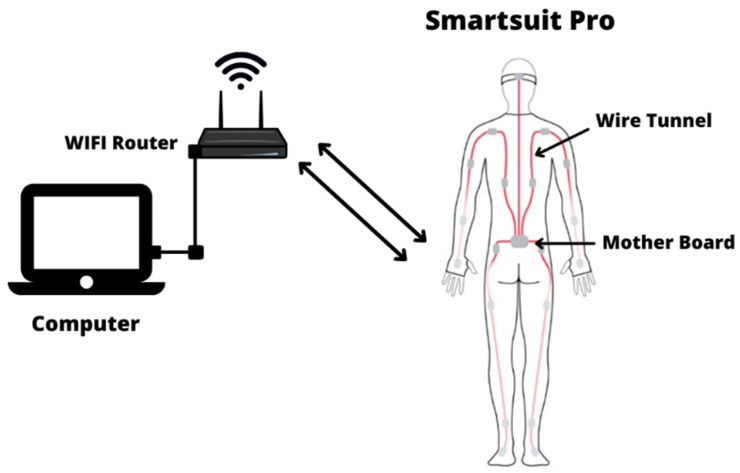
Rokoko Smartsuit Pro sensor placement and hardware setup.

**Figure 3 ijerph-19-14232-f003:**
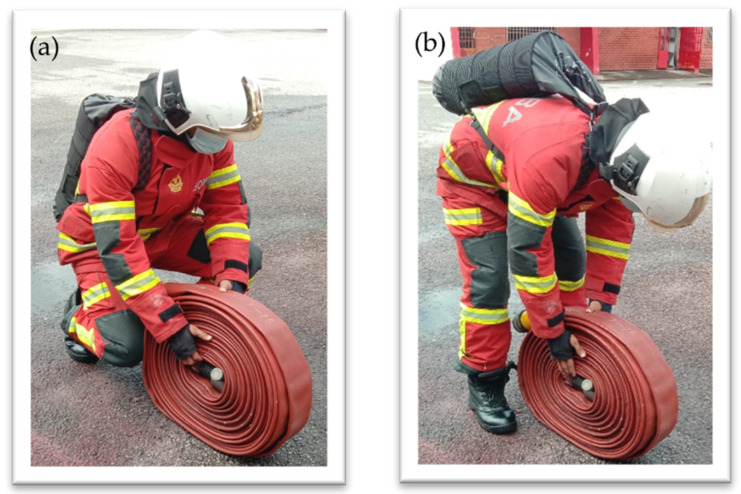
Type of ground lifting posture observed. (**a**) Squatting and (**b**) stooping.

**Figure 4 ijerph-19-14232-f004:**
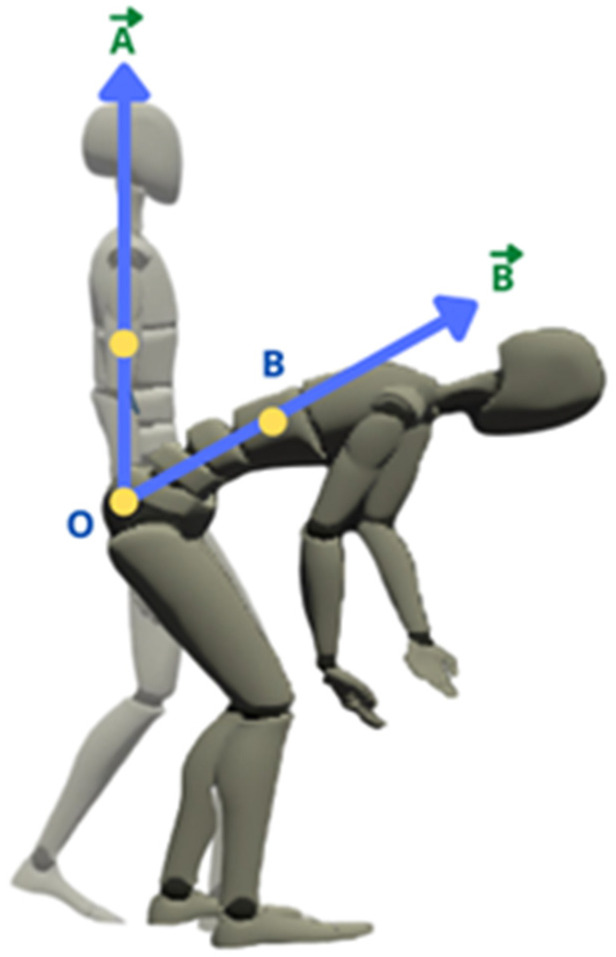
The schematic diagram for the definition of vector for thorax extension.

**Figure 5 ijerph-19-14232-f005:**
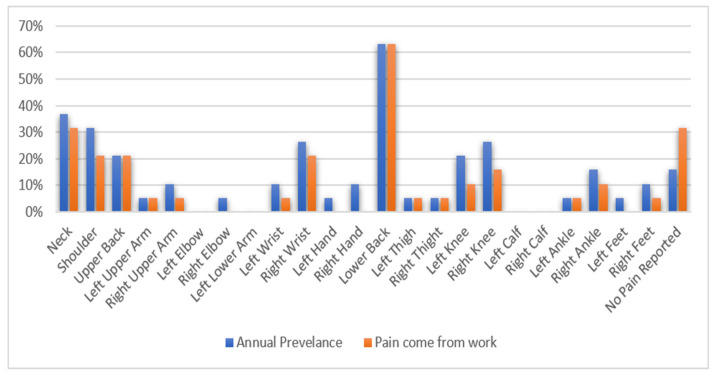
Percentage of respondents experiencing pain in any region of the body in the past 12 months and attributing pain to work.

**Figure 6 ijerph-19-14232-f006:**
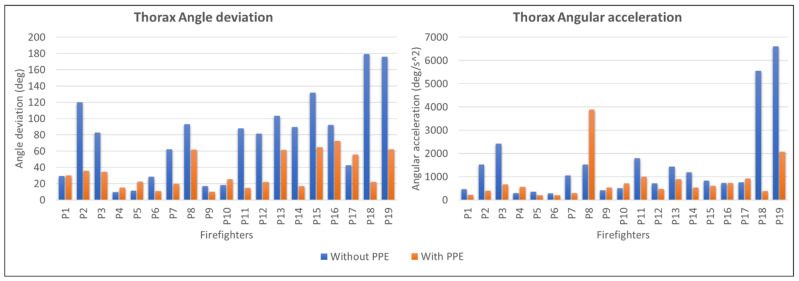
Thorax angle deviation and thorax angular acceleration during lifting.

**Figure 7 ijerph-19-14232-f007:**
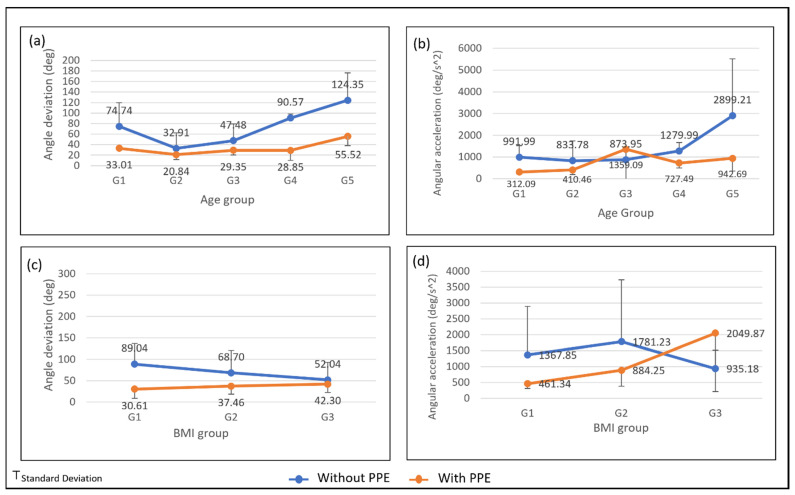
The average and standard deviation of thorax extension for (**a**) angle deviation by age group; (**b**) angular acceleration by age group; (**c**) angle deviation by BMI group; and (**d**) angular acceleration by BMI group.

**Figure 8 ijerph-19-14232-f008:**
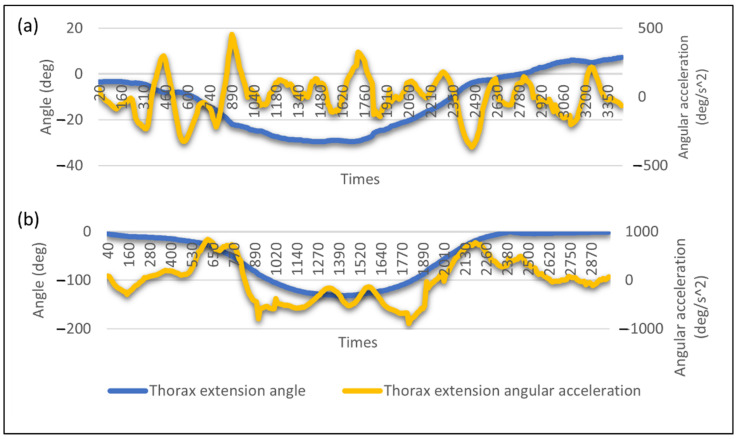
Example of angle deviation and angular acceleration profile for (**a**) squatting lifting posture and (**b**) stooping lifting posture.

**Figure 9 ijerph-19-14232-f009:**
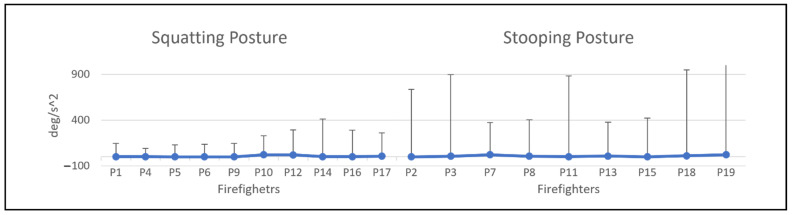
Average and standard deviation of the angular acceleration of FFs for squatting and stooping posture during lifting.

**Figure 10 ijerph-19-14232-f010:**
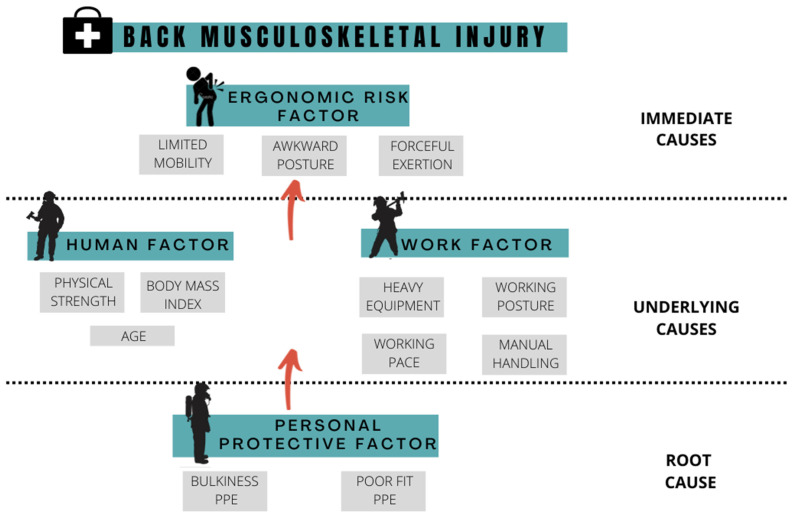
The framework of causes for back musculoskeletal injury of FFs.

**Table 1 ijerph-19-14232-t001:** FFs demographic background.

Age Group	Participant	Age	Service Experienced	Height	Weight	BMI
G1	P1	24	2	168	74	26.2
	P2	25	6	176	77	24.9
G2	P3	29	5	169	78	27.3
	P4	26	2	169	80	28.0
	P5	28	10	170	88	30.4
	P6	29	2	179	63	19.7
G3	P7	34	10	167	65	23.3
	P8	35	15	167	88	31.6
	P9	31	3	170	64	22.1
	P10	34	10	167	78	28.0
G4	P11	37	17	181	94	28.7
	P12	39	10	175	64	20.9
	P13	39	16	168	78	27.6
	P14	36	7	165	56	20.6
G5	P15	43	20	173	62	20.7
	P16	43	20	176	75	24.2
	P17	47	15	157	65	26.4
	P18	46	21	175	60	19.6
	P19	53	28	173	78	26.1

**Table 2 ijerph-19-14232-t002:** Comparison statistical test.

Data Type	*p* Value				
	*t*-Test	ANOVA			
	Condition with PPE and without PPE	Age Group		BMI Group	
		without PPE	with PPE	without PPE	with PPE
Angle deviation	0.001	0.110	0.045	0.339	0.545
Angular acceleration	0.089	0.344	0.546	0.047	0.805

## Data Availability

Not applicable.

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
