# Peer review of "Biomechanics Analysis of the Firefighters’ Thorax Movement on Personal Protective Equipment during Lifting Task Using Inertial Measurement Unit Motion Capture"

_ijerph, 2022, doi:10.3390/ijerph192114232_

Round 1
Reviewer 1 Report (Previous Reviewer 2)
Dear Authors, you edited a lot in your paper. Unfortunately I detected more small typos, you introduced errors by your edits. You un-improved your paper.
Unfortunately you did not give proof of your results, by statistical tests. It is not good to only give verbal discussions without results. Instead of proof you choose to design a Discussion (Chapter 3) of length of about 30-40% of your whole paper. Your Results (= your second Chapter 3) belongs to the Chapter 2 Data Analysis.
The IJERPH is a high impact journal. You should prove your observations by tests. I do think that your current version is too erroneous for this journal. I suggest the Editor to reject it.
To sustain my point of view I give details and critiques about - what I see - as severe errors and badly reporting.
Details, typo’s, suggestions for improvements and discussions
1. Line 2, ‘the Firefighter’s’ should be ‘Firefighters’.
2. Line 17, announces a set of questionnaires, as used by you. What set is it? You have no reference to this set of questionnaires, and it is not in the files of Additional material for the paper. Is it a workbench of standardized questionnaires (for this type of research)?
3. Line 20, announces ‘the’ descriptive analysis to analyse the data. The descriptive analysis is not yet available in your paper. The analysis in this paper is confined to four lines from 537 to 540.
4. Lines 23-24, you conclude ‘… biomechanics analysis of thorax angle deviation and angular acceleration has shown that using FFs PPE significantly causes restricted movement and limited mobility for the FFs.’ Is this from the literature, or your opinion?
5. Lines 25-26, your results announced here are unsustained: the relation between age, BMI and back pain.
6. Line 158, it is incorrect to start a sentence by a reference: ‘[21,22]’.
7. Lines 168-169, you say 'both motion capture has ...'. This is wrong: both motion captures has to be plural.
8. Line 174, you do not give references of the many studies extensively conducted.
9. Lines 175, 176 and 177, say three times in different words what your study and its subsequent report (this paper) entails. This is redundant. Please merge these three sentences together such that you do not need to say three times again ‘this study’.
10. Line 229, you say 'The participant for this study'. This should read 'Subjects of this study' or 'Participants of this study'. You do have more than one participating subject. So "The" participant does not exist.
11. Lines 233 – 235. You have split the sample of size N of this study and calculated the minima of the Age groups apparently by the formula.(1).
What is missing here in your text is: what do the variables in the formula mean? What is z here and the P estimator? And the epsilon function of the population size? Your formula is close to the formula in this retracted paper [1]. As it stands now in your text is your formula unrelated to your data.
12. Lines 238-239, you mention 'The musculoskeletal discomfort survey' without reference. You refer to this survey totally six times. Please put in a reference to it. Is it your reference [4]?
13. Between Lines 407-408, you mix up notation of the formulas 4, 5 and 6. In (5) you abusively use an outproduct notation. The inproduct notation in the denominator of (6) is not available/explained in (4) and (5). Please improve your notation, it is wrong and misleading.
14. Line 531, you use here radians for the angles. This is wrong. Your promised that all angles will be in degrees, in Line 404 you promise that angles will be measured in degrees.
15. Line 546, Table 1, “Group age” should read “Age Group”. This error happens in three places in the text.
16. Lines 536-540 report your data analysis. It misses here the test of your grouping of participants/subjects into age groups. Is this grouping sustained? What you did is computing a-priori the size of the age groups. This is not a test of the correctness, but an estimator to size.
A test for your grouping is the Chi-squared (= Chi2) test. You estimated group size by a (seemingly) wrong formula, see above item 11. The Chi2 is needed to test if your grouping is unbiased. You find it very well explained with examples in [2], Chapter 10.2 ‘Chi-Square tests’.
17. Lines 543-563, do not belong to Results. Grouping of your data belongs to your Data Analysis chapter.
18. Between Lines 598-599, in Figure 5 you have erroneously ‘de derivation’ vertically at the left.
19. Between Lines 598-599, you erroneously mix degrees and radians in the same Figure 5.
20. Lines 602-1213, gives your discussion. It is too long. Where are your results?
21. Between Lines 800-801, you have put here your illustration in Figure 6. Isn’t this late in the report? Shouldn’t this be placed close to Figure 2?
22. Lines 1319 and 1321 give twice the same formulation ‘will help’. One of these is redundant.
23. Lines 1318-1324, recommend to standardize the ideal acceleration values and angle deviations during lifting tasks. Aren’t these standards available in text books? Or in published papers? Research to find the ideal, i.e. to standardize back and other movements is done for robotics, in [3].
24. Line 1321, what is ‘ss’?
25. Lines 1330 and 1332 both have the initials of the first author wrong. It should read ‘M.N.H.Y.’ stead of currently ‘M.H.N.Y.’ and ‘MH.Y’. The same happens to the co-author N.A., he should read: N.Z.A.
26. Line 1515, the page numbers are incorrect.
Author Response
Thank you for the extensive and beneficial comments on the improvement of our manuscript.
Please see the attachment for the changes made per the comments.
Thank you again.

Reviewer 2 Report (Previous Reviewer 3)
The paper is vThey should just check all the abbreviations.
It should also be more exact in developing the methodology.
THank you for your contribution to the scientific community.
Author Response
Thank you for your positive comments. The authors have reviewed the abbreviations for the manuscript.
Thank you again.
Round 2
Reviewer 1 Report (Previous Reviewer 2)
Dear Authors, you did a lot of work on the text.
I still have some doubt about the quality of the end product. For instance:
In line 382 lifting is expressed in degrees and in line 386 is lifting expressed in radians. This is inconsistent.
Example: I do not find use of the output of the statistical test in Table 2.
I think the paper has merits and is 'just passing' in present form.
This manuscript is a resubmission of an earlier submission. The following is a list of the peer review reports and author responses from that submission.
Round 1
Reviewer 1 Report
Dear authors,
The manuscript has some issues that must be addressed accordingly.
Please, make sure all my comments will be replied in a point-to-point cover letter.
The comments are in the attached file.

Reviewer 2 Report
Dear Authors, your paper is well designed and some parts are clearly written. My compliments. As with all good works, there are flaws (typos and missing text) see details in the list below.
It might feel hopeless for you, so many remarks. But remind: it is my intention to help you with details to improve your paper, stead of the usual very brief remarks from referees, letting you alone with your text. Moreover, the MDPI Editors belong to the best of the world, they will point in a later stage to typos. So, it is easy for you if I list a number of them in this early stage. I talk to you with over 50 years of experience in scientific research, so my judgment about publishers is sustained.
The main issues in the paper are:
A. Statistical tests to sustain your results are missing;
B The medical (and critical) remark about the proven relation between back pain and obesity is missing;
C. You have misplaced Data Analysis in the Results and Discussion chapter;
Details, typos, suggestions for improvements, and discussions
1. In the text you have 128 times the full word spelled out: firefighter, firefighters, firefighter's. Isn’t it better for the reader to use an acronym, for instance, FF, FFs, etc.?
2. Line 52, ‘develops’ should read ‘develop’.
3. Line 57, ‘Kinematic’ should read ‘Kinematics’.
4. Line 63, the word ‘in’ has to be deleted here.
5. Line 68, ‘can be divided into optical motion capture and non-optical motion 68 capture [26].’ could be improved.
6. Line 97, the reference to the survey ‘musculosketal’ is missing!
7. Line 101, ‘the study’ should read ‘this study’, or ‘current study’, or ‘the study reported here’.
8. Line 102, a reference to the Rokoko suit is missing. What is the manufacturer, and has it been used and tested at other research institutes?
9. Line 111, here starts the use of the term ‘simulation’. Shouldn’t this be an ‘experiment’? If I am incorrect, then please explain the term: simulation here, because the behavior of the FFs is the real thing in the experiment for this paper, I assume.
10. Between Lines 115 – 116, the figure seems unrelated to the description in the text. Where in figure 2 is the hub from Line 112? The figure displays a computer: what does it do?
11. Line 114, where is the tunnel in Figure 2?
12. Line 115, should the term ‘simulation’ be replaced here by ‘experiment’? See also Item 9 above.
13. Line 121, what is ‘fps’? Is it files per second? If you mean frequency here, it is ‘Hz’ (from the German scientist Herz).
14. Lines 123 - 124, talk about a simulation, avoiding magnetic fields, etc. A description of the ‘simulation’ is missing; in many places (for example line 193) you talk about a simulation. Is it the research experiment itself?
15. Lines 129 - 130, the sentence ‘The selected task is fire hose ground lifting which required 129 the firefighter to perform thorax extension to complete the task.’ Seems to be verbose. I think it is better to say ‘The selected fire hose ground lifting task requires thorax extension performance.
16. Line 137, ‘In the biomechanics study of human movement, there are 4 main movement types:’ could be improved by ‘Biomechanically human movement has four main types:’.
17. Line 138, ‘This study will focus on the angle deviation and angular acceleration of the thorax extension of firefighters, as shown in figure 3.’ should be placed in Line 141 after the sentence ending with [34].
18. Line 145, ‘Rokoko motion capture created a line of the vector …’ should read ‘The Rokoko motion capture creates a line of the vector …’.
19. Line 145 - 163, here you explain the Rokoko data collection. This should belong to previous section: ‘2.1 Data Collection’.
20. Line 158, it is customary to denote variables italic as you correctly do in formula (9). But units such as radian and seconds are not written italic. Also is the italic acceleration inconsistent with angular?
21. Lines 160 - 163, you use the future tense here ‘will discuss…’, ‘will describe …’. Isn’t it better to just say: ‘uses’, describes’?
22. From Line 164, in the Results and Discussion chapter, I miss the results of medical research indicating that high BMI is related to back pain. The majority of the subjects in your study have a high BMI, so you should relate their complaint with this medical cause. Research indicating this relation is in [1–8].
23. Line 166, the percentage sign is missing after 37; and ‘low’ in the text should be: less.
24. Line 166 - 167, delete the words ‘, and 63% had more than 10 years of service experience‘ because the remark is redundant. You just have said complimentary that 37% of the FFs have less than 10 years of experience. The reader shouldn’t be treated if she/is a nitwit.
25. Line 169, I recommend replacing ‘as shown’ by ‘and shown’.
26. Line 172, to get rid of one of the many repetitions of ‘firefighters’, please replace ‘all the FFs that reported lower back pain’, with ‘all subjects reported lower back pain’.
27. Line 179, replace ‘will focus’ with ‘focused’.
28. Line 193, 'have been observed performed’ is wrong. Please change it to ‘’have been performed’.
29. Between Lines 225 - 226, the vertical word ‘angular’ has inconsistently a and capital A.
30. Between Lines 246 - 312, I think that these current text lines on Data Analysis should be moved to Data Collection. In the analysis belongs a statistical test.
31. Line 250, the sentence could be improved by ‘16% of FFs show a minimally increasing …’. Then you can delete the next sentence ‘However, the increasing angle is minimal.’
32. Lines 256 – 257, give a conclusion sentence ‘It belongs . . . injury.’ This should be moved to the Conclusions chapter.
33. Line 262, suddenly you use degrees here, while before you use radians. Please be consistent.
34. Between Lines 322 – 323, in Figure 10 is BMI missing in the Human Factor column. See my remark 3 before.
35. Line 325, here appears suddenly ‘personal protective equipment'. This is inconsistent with your previous use of the short-cut PPE. You could also improve the opening of the sentence by saying ‘The PPE causing bulkiness and poor fit will affect …’
36. Line 332, the words ‘factor from’ should read ‘part of’, or something else. You should avoid your current expression: ‘factor from factor’.
37. Line 346, ‘discovered is wrong here. It should read ‘revealed’ or ‘showed’.
38. Line 350, ‘results have shown that data on body angle and’ could be improved by ‘results show that body angle and’.
39. Line 359, ‘The ergonomic intervention to the human’ should read ‘Ergonomic intervention of human’.
40. Lines 377 – 481, many references have erroneous page numbers. For instance, in [9] you only refer to the page count, not to the page numbers. This page numbering error happens more often. (https://www.mendeley.com/guides/using-citation-editor/) to manage your references. It is provably the best ref manager.
41. Lines 377 – 481, at least eight references do not have page numbers. Please use Mendeley.
42. Lines 377 – 481, many references do not end with a dot.
References
1. Leigh, J.P.; Sheetz, R.M. Prevalence of back pain among fulltime United States workers. Br. J. of industrial Med. 1989, 46, 651–657.
2. Shiri, R.; Karppinen, J.; Leino-Arjas, P.; Solovieva, S.; Viikari-Juntura, E. The Association Between Obesity and Low Back Pain: A Meta-Analysis. Am. J. Epidemiol. 2010, 171, 135–154.
3. Dario, A.B.; Ferreira, M.L.; Refshauge, K.M.; Lima, T.S.; Ordoñana, J.R.; Ferreira, P.H. The relationship between obesity, low back pain, and lumbar disc degeneration when genetics and the environment are considered: a systematic review of twin studies. Spine J. 2015, 15, 1106–1117.
4. Koyanagi, A.; Stickley, A.; Garin, N.; Miret, M.; Ayuso-Mateos, J.L.; Leonardi, M.; Koskinen, S.; Galas, A.; Haro, J.M. The association between obesity and back pain in nine countries: A cross-sectional study. BMC Public Health 2015, 15, 1–9.
5. Chou, L.; Brady, S.R.E.; Urquhart, D.M.; Teichtahl, A.J.; Cicuttini, F.M.; Pasco, J.A.; Brennan-Olsen, S.L.; Wluka, A.E. The Association Between Obesity and Low Back Pain and Disability Is Affected by Mood Disorders: A Population-Based, Cross-Sectional Study of Men. Medicine (Baltimore). 2016, 95.
6. Zhang, T.T.; Liu, Z.; Liu, Y.L.; Zhao, J.J.; Liu, D.W.; Tian, Q.B. Obesity as a Risk Factor for Low Back Pain. Clin. Spine Surg. 2018, 31, 22–27.
7. Šagát, P.; Bartík, P.; González, P.P.; Tohănean, D.I.; Knjaz, D. Impact of COVID-19 Quarantine on Low Back Pain Intensity, Prevalence, and Associated Risk Factors among Adult Citizens Residing in Riyadh (Saudi Arabia): A Cross-Sectional Study. Int. J. Environ. Res. Public Health 2020, 17, 7302: 1–13.
8. Merkus, S.L.; Coenen, P.; Forsman, M.; Knardahl, S.; Veiersted, K.B.; Mathiassen, S.E. An Exploratory Study on the Physical Activity Health Paradox—Musculoskeletal Pain and Cardiovascular Load during Work and Leisure in Construction and Healthcare Workers. Int. J. Environ. Res. Public Heal. 2022, Vol. 19, Page 2751 2022, 19, 2751.
9. Kim, M.G.; Kim, K.-S.; Ryoo, J.-H.; Yoo, S.-W. Relationship between Occupational Stress and Work-related Musculoskeletal Disorders in Korean Male Firefighters. Ann. Occup. Environ. Med. 2013, 25, 1–9.
Reviewer 3 Report
The article is very interesting.
I would like the authors to revise the abbreviations in the title and describe the method. Since the abbreviations have not been explained, it may cause doubt to a reader (it has been revealed only in the Introduction what IMU stands for and no abbreviations should appear in the title).
The Authors should describe the limitations of the study.